# Effects of 0.01% Atropine Instillation Assessed Using Swept-Source Anterior Segment Optical Coherence Tomography

**DOI:** 10.3390/jcm10194384

**Published:** 2021-09-25

**Authors:** Tadahiro Mitsukawa, Yumi Suzuki, Yosuke Momota, Shun Suzuki, Masakazu Yamada

**Affiliations:** Department of Ophthalmology, Kyorin University School of Medicine, 6-20-2 Shinkawa, Mitaka, Tokyo 181-8611, Japan; mitsukawa@ks.kyorin-u.ac.jp (T.M.); momota@ks.kyorin-u.ac.jp (Y.M.); suzuki-s@ks.kyorin-u.ac.jp (S.S.); yamadamasakazu@ks.kyorin-u.ac.jp (M.Y.)

**Keywords:** accommodation, anterior segment optical coherence tomography, low-concentration atropine, myopia, ocular biometric components

## Abstract

In this paper, we assessed the short-term effects of 0.01% atropine eye drops on anterior segment parameters by performing ocular biometry using a swept-source anterior segment optical coherence tomography system (AS-OCT). We recruited 17 healthy volunteers (10 men and 7 women aged 24–35 years) with no history of eye disease. Participants without accommodative demand demonstrated significant mydriasis 1 h after the atropine instillation (4.58 ± 0.77 to 5.41 ± 0.83 mm). Pupil diameters with a 5 diopter (D) accommodative stimulus at 1 h (4.70 ± 1.13 mm) and 24 h (4.05 ± 1.06 mm) after atropine instillation were significantly larger than those at baseline (3.71 ± 0.84 mm). Barring pupil diameter, no other biometric parameters significantly changed at any point in time after atropine instillation without accommodative demand. However, with an accommodative stimulus, anterior chamber depth (ACD) at 1 h and posterior curvature of the lens at 1 and 24 h were both significantly larger than those before atropine instillation. Using AS-OCT, we detected a slight decrease in the accommodation response of ocular biometric components evoked by 0.01% atropine instillation. Morphologically, our measurements suggested a change in the ACD and horizontal radius of the lens’ posterior surface curvatures due to the subtle reduction of accommodation.

## 1. Introduction

Myopia is the leading cause of preventable visual impairment in childhood and adolescence [1,2]. An increasing prevalence of myopia has been reported in East and Southeast Asia, including China, Korea, and Japan [1,2,3,4,5,6]. In addition, the number of patients with myopia has increased in the United States and Europe, mainly among school-aged children and young adults [2,7,8]. As a result, the global prevalence of myopia, including pathologic myopia, is increasing, and has gained prominent attention as a social health problem. Complications resulting from myopia can incur large social and economic costs [9]. Therefore, the prevention of myopia progression has become increasingly important.

Myopia is generally present at the school-going age in patients. However, with the use of appropriate treatment modalities targeting children with myopia, it is possible to reduce the lifetime risk of retinal complications by reducing the severity of final myopia [10]. Several methods for the prevention of myopia progression have been reported to date, and they are broadly classified into nonpharmacological and pharmacological treatments. The former includes optical approaches such as the use of special spectacles, contact lenses, and orthokeratology [11,12,13,14,15]. The latter relies on the use of atropine eye drops, which are an established pharmacotherapy for the prevention of myopia progression [16,17,18].

Owing to its antimuscarinic action, atropine has long been used in ophthalmology in the form of 1% atropine eye drops for accommodation paralysis, and as an anti-inflammatory agent for conditions such as keratitis and iritis [19]. A study in 2006, Atropine for the Treatment of Myopia (ATOM-1), reported that the use of 1% atropine was effective in halting the progression of myopia [16]. However, over a 2 year period, the researchers observed photophobia resulting from dilated pupils and impaired near vision due to accommodation paralysis in eyes treated with 1% atropine. These side effects greatly interfered with the daily lives of patients. In addition to the rapid progression of myopia after discontinuation of the eye drops, a 1% concentration was considered inappropriate for myopia control [20]. As a result of this, that same research group subsequently conducted a study using various low-concentration atropine treatments. In 2012, they reported the results of a clinical study that assessed the inhibitory effect of atropine on myopia progression (ATOM-2), using 0.5%, 0.1%, and 0.01% atropine eye drops [17]. In their survey of the prevention of 2 year myopia progression, the researchers found that the group that received the lowest concentration of atropine (0.01%) achieved approximately half the inhibitory effect of the placebo group (−0.49 diopters (D) compared with −1.20 D). Furthermore, the instillation of 0.01% atropine resulted in minimal adverse reactions when compared with the instillation of 0.1% and 0.5% atropine [17]. Consequently, the use of low-concentration atropine to reduce myopia progression has garnered attention because of its limited effect on visual function. However, in the ATOM-2 study, Chia et al. [17] reported that a small proportion (6%) of the patients required combined photochromatic progressive glasses because they developed impaired near vision and photophobia. In a study of the use of low-concentration atropine for preventing myopia progression, Yam et al. [18] assessed patients using a visual function questionnaire and found that atropine instillation had no effect on general vision, near vision activities, social functioning, or color vision. Although the instillation of 0.01% atropine eye drops only has a subtle effect on the pupil diameter and accommodative amplitude, concerns remain regarding the undesirable effects of atropine on patients’ daily life [17].

Nonetheless, there are limited data regarding the short-term effects of low-concentration atropine instillation on pupil diameter and accommodative function in young adult subjects [21]. In our previous study, we successfully analyzed ocular biometric components (OBCs), including changes in the crystalline lens during accommodation, and the effects of cycloplegics, using a commercially available anterior segment optical coherence tomography (AS-OCT) system [22]. In recent years, AS-OCT has been used for in vivo studies of ocular lens behavior during accommodation. A newly developed swept-source AS-OCT system (CASIA 2, Tomey Corp., Nagoya, Japan) has enabled detailed biometry measurements to be obtained from the corneal surface to the posterior surface of the lens by elongating the range of the imaging depth and increasing the sensitivity [23,24]. In the present study, we used the AS-OCT system to quantitatively evaluate the effects of 0.01% atropine eye drops on OBCs in the anterior segment of the eye.

The current study aims to determine how the instillation of 0.01% atropine produces morphological changes in the eye by assessing ocular biometric components (OBCs) before and after instillation, using anterior-segment optical coherence tomography. 

## 2. Subjects and Methods

### 2.1. Participants

This study followed the guidelines outlined in the Declaration of Helsinki from the World Medical Association. All participants received a full explanation of the procedures and they provided written informed consent before they agreed to participate in the study. The study protocol was approved by the Institutional Review Board of Kyorin University School of Medicine (Project H30-099).

In this study, we examined young adults rather than children, as low-concentration atropine eyedrops for myopia have not been approved for children in Japan. 

The study participants included 17 healthy volunteers (10 men and 7 women) aged 24–35 years (mean ± standard deviation: 28.9 ± 3.6 years). None of the participants had a history of eye disease, except for refractive errors, and all had a best-corrected visual acuity of 20/20 or better. The exclusion criteria were a history of any ocular disease, ophthalmic surgery, or laser treatment. We also excluded participants who were taking systemic medications that could affect accommodation.

We examined each participant’s noncycloplegic refraction using an ARK-1 autorefractor (NIDEK Co. Ltd., Gamagori, Japan). We considered the effect that the degree of refractive error would give to accommodation factors, as 17 participants had refractive errors from approximately -11 D to 0 D [25]. However, in this study, to ensure participants’ ability to accommodate 5 D or greater, we also examined the accommodation of the participants using the ARK-1 autorefractor.

### 2.2. Procedures and Assessments

We examined both eyes of all participants using the CASIA 2 swept-source AS-OCT system. The AS-OCT device has a swept-source laser that operates at a central wavelength of 1310 nm and a scan rate of 50,000 A-scans per second. The maximum imaging area is 16.0 mm × 16.0 mm, and the maximum imaging depth is 11.0 mm. This device enables simultaneous biometry measurements to be obtained for all anterior segment structures, including the cornea, anterior chamber, and crystalline lens.

All OCT images were obtained in a dimly lit examination room. During the measurements, the participants were instructed to fixate on the coaxial accommodative target image present in the OCT device. The negative or positive lens was set to compensate for the participant’s spherical ametropia for near-equivalent spherical refractive correction. Next, we added a −5.0 D lens to stimulate physiological accommodation using an optical system in the OCT system. The active eye tracker of the OCT system was centered on the participant’s eye. Two experienced operators (M.Y. and S.S.) collected all images.

Measurements were performed with and without a single instillation of 0.01% atropine eye drops. To prepare the 0.01% atropine eye drops, commercial 1% atropine sulfate hydrate (Nitten ATROPINE Ophthalmic Solution 1%; Nitten Pharmaceutical Co. Ltd., Nagoya, Japan) was diluted with saline. OCT images of the eye were obtained before instillation, and at 1, 24, and 48 h after instillation. The OBCs measured using AS-OCT included pupil diameter, anterior chamber depth (ACD), lens thickness (LT), and the horizontal radii of the lens’ anterior curvature (LAC) and lens’ posterior surface curvature (LPC). The boundaries of both the cornea and lens were outlined for anterior segment biometry. The positioning of the anterior and posterior surfaces of the lens on the horizontal meridian was traced, and the radius of the crystalline lens was determined using measurements that permitted circular fitting to the anterior and posterior lens surfaces.

The participants’ accommodative amplitude was measured using the ARK-1 autorefractor before instillation, and at 1, 24, and 48 h after instillation of the 0.01% atropine eye drops. 

Objective measurement of accommodation was performed with the participant focusing on a target that moved to a near point from a distance. Additionally, we conducted the measurement of participants’ axial length using an optical axial length measuring device (OPTICAL BIOMETER OA-2000, Tomey Corp., Nagoya, Japan)

The participants were also instructed to answer questionnaires 1, 24, and 48 h after atropine administration about the difficulties they experienced with near vision and photophobia, separately, in which they rated their symptoms on a scale ranging from 0 (none) to 10 (inability to perform daily tasks).

### 2.3. Statistical Analysis

The Statistical Package for the Social Sciences version 27.0 for Windows (IBM Armonk, NY, USA) was used for all statistical analyses. The Mann–Whitney U test and Wilcoxon signed-rank test were used to perform comparisons. *p*-Values < 0.05 were considered to indicate statistical significance.

## 3. Results

### 3.1. Baseline Characteristics of the Participants

Table 1 presents the baseline biometric parameters of both eyes before the instillation of 0.01% atropine eye drops. The noncycloplegic refraction of the right eye ranged from −0.38 to −10.88 D, and that of the left eye ranged from +0.38 to −11.25 D; there was no significant difference in refraction between the right and left eyes (*p* = 0.691, Mann–Whitney U test). There was also no significant difference in accommodation between the right and left eyes, and both eyes were able to accommodate more than 5 D. We did not find any significant differences in any baseline biometric parameters between the right and left eyes before the instillation of 0.01% atropine eye drops (Mann–Whitney U test). Therefore, we present the findings of only the right eyes.

### 3.2. Effects of 0.01% Atropine on Pupil Diameter

Figure 1a presents a comparison of pupil diameters in the relaxed state, and those with the 5 D accommodative stimulus, before and 1, 24, and 48 h after instillation of 0.01% atropine eye drops. The pupil diameter was significantly larger 1 h after atropine instillation than before the atropine instillation (from 4.58 ± 0.77 to 5.41 ± 0.83 mm) in the relaxed state (*p* < 0.05, Wilcoxon signed-rank test). With the 5 D accommodative stimulus, the pupil diameter at 1 and 24 h was significantly larger than that before atropine instillation (from 3.71 ± 0.84 mm to 4.70 ± 1.13 and 4.05 ± 1.06 mm, respectively, *p* < 0.05, Wilcoxon signed-rank test). In contrast, there was no significant difference in the pupil diameter 24 and 48 h after atropine instillation compared with that before atropine instillation in the relaxed state, or in the pupil diameter 48 h after atropine instillation compared to before atropine instillation with the 5 D accommodative stimulus.

### 3.3. Effects of Atropine on Other Biometric Parameters Measured Using AS-OCT

Figure 1b–e shows a comparison of biometric parameters (ACD, LT, LAC, LPC) between before and 1, 24, and 48 h after the instillation of 0.01% atropine eye drops in the relaxed state and with the 5 D accommodative stimulus. Other than pupil diameter, none of the biometric parameters showed changes in the relaxed state at any point in time when compared to before the instillation.

However, with the 5 D accommodative stimulus, ACD at 1 h was significantly larger than that before the instillation (from 3.08 ± 0.16 mm to 3.10 ± 0.18 mm, *p* < 0.05). LPC at 1 and 24 h was significantly larger than that before the instillation (from 5.21 ± 0.43 mm to 5.36 ± 0.35 and 5.50 ± 0.50 mm, respectively, *p* < 0.05).

### 3.4. Effects of Atropine on Refraction, Accommodation Amplitude, and Subjective Symptoms

Table 2 shows the spherical equivalent 1, 24, and 48 h after the instillation of 0.01% atropine eye drops. There were no significant changes in the mean spherical equivalent from the values before the instillation (*p* = 0.10, *p* = 0.86, and *p* = 0.55, respectively).

Table 2 shows the accommodative amplitudes 1, 24, and 48 h after the instillation of 0.01% atropine eye drops. There were no significant changes in the mean accommodative amplitude at any point in time when compared with that before the instillation (*p* = 0.76, *p* = 0.50, and *p* = 0.07, respectively).

In terms of the two subjective symptoms, we found no serious adverse events related to atropine. None of the participants reported photophobic sensation, although three participants reported mild difficulty with near vision (rated as 1/10 and 3/10 in one and two participants, respectively) 1 h after the atropine instillation.

## 4. Discussion

Recent studies have shown that atropine effectively inhibits the progression of myopia and axial elongation [16,17,18,26,27]. Treatment guidelines for the inhibition of myopia progression, developed by Wu et al. [27], ranked low-concentration atropine eye drops as the key component to successful inhibition. The reported side effects of low-concentration atropine eye drops were limited to photophobia due to mydriasis and impaired near vision resulting from the impairment of accommodative amplitude [17,18,21]. Although these adverse events were rare and mild, objective measurements of changes in OBCs after 0.01% atropine instillation might be important. With this background in mind, we assessed the effects of low-concentration atropine eye drops on OBCs using AS-OCT. Our results revealed significant but subtle changes in OBCs.

In our previous study, we used a commercially available AS-OCT system (CASIA 2) to measure the OBCs, including lens parameters [22]. This system enables detailed biometric measurements to be obtained from the corneal surface to the posterior lens surface by increasing the range of the imaging depth and improving performance sensitivity [23,24]. Our prior study revealed an increase in LT and a decrease in ACD, LAC, and LPC with accommodation, which suggested that steepening and anterior movement of the lens during accommodation occurred. After the application of cycloplegics (cyclopentolate), there was a decrease in LT, which resulted in an equivalent increase in ACD [22]. Therefore, the CASIA 2 swept-source AS-OCT system could detect changes in OBCs during accommodation.

Accordingly, we used the same technique in this study to assess OBCs before and 1, 24, and 48 h after the instillation of 0.01% atropine. Although no participants reported photophobic sensations, 0.01% atropine had a minor effect on pupil diameter. While the pupil diameter increased significantly 1 h after instillation in a relaxed pupil state, it returned to the pre-instillation level at 24 h. The pupil diameters at 1 and 24 h were significantly larger with a 5 D accommodative stimulus, but they returned to the pre-instillation level at 48 h. Kaymak et al. [21] reported the short-term effects of 0.01% atropine instillation on pupil diameter and accommodation amplitude in 14 young adults. The reported pupil diameters before and 24 h after instillation were 3.3 ± 0.5 and 3.9 ± 0.8 mm, respectively, which indicated a significant increase (*p* < 0.02). Our study also confirmed that instillation of 0.01% atropine caused a slight and transient increase in pupil diameter.

In the relaxed state, none of the assessed OBCs (ACD, LT, LAC, and LPC), other than pupil diameter, showed significant changes at any of the assessed points in time compared to before the instillation. In contrast, with the 5 D accommodative stimulus, ACD 1 h and LPC 1 and 24 h after atropine instillation were significantly larger than those before the treatment (*p* < 0.05). However, there were no differences in either LT or LAC. Therefore, we confirmed that the cycloplegic effect following the instillation of 0.01% atropine eye drops was marginal. Our results suggest that measuring OBCs using the AS-OCT system is useful for detecting subtle changes that result from low-concentration atropine instillation. The AS-OCT results corresponded with the measurement of the accommodation amplitude. In our study, we found no decrease in the accommodation amplitude as a result of the instillation of 0.01% atropine. Only a few participants reported experiencing some difficulties with near vision 1 h after atropine instillation. Similarly, Kaymak et al. [21] reported no difference in the accommodation amplitude before and 24 h after 0.01% atropine instillation (*p* = 0.06).

Our study has some limitations. First, the participants were young adults rather than school-aged children, which could have influenced the results. Low-concentration atropine instillation has been used to inhibit myopia progression in school- and preschool-aged children to address the trend of early-onset myopia and the increase in the number of preschool- and school-aged patients with myopia. In this respect, the ocular permeability of atropine and its pharmacokinetics might differ between children and adults. The accommodation amplitude also differs between school-aged children and young adults. Hence, as the participants were young adults aged 24‒35 years, the results might not be directly applicable to school-aged children. Second, we observed only the short-term effects of a single instillation of 0.01% atropine. Our study showed that at a dose of 0.01% atropine, short-term effects included a slight increase in pupil diameter and minor accommodation paralysis. However, the long-term effects of low-concentration atropine instillation are not clear, and further studies are needed to clarify this issue. Third, similar to most other studies using AS-OCT, we were unable to analyze the entire lens shape through the pupil [28,29,30,31,32,33]. Because of the variability in the measurements and the asphericity of the lens, the curvature radius obtained by fitting the circular curve might not precisely express the shape of the lens. Finally, although the effect of atropine eye drops on vergence reactions should have been evaluated, we did not examine this in the present study.

In conclusion, we assessed the effects of 0.01% atropine eye drops by performing ocular biometry using the CASIA 2 AS-OCT system. Similar to the findings in previous reports, we did not observe significant photophobia or subjective difficulty in near vision. However, our measurements did suggest a change in the pupil diameter, ACD, and LPC, which are part of the assessed OBCs, which resulted from a subtle reduction in accommodation. In other words, morphologically, we were able to confirm an increase in the pupil diameter and a decrease in the accommodation response of OBCs with a 5 D accommodative stimulus following the instillation of 0.01% atropine. Moreover, we demonstrated that AS-OCT could evaluate subtle changes evoked by low-concentration atropine administration.

## Figures and Tables

**Figure 1 jcm-10-04384-f001:**
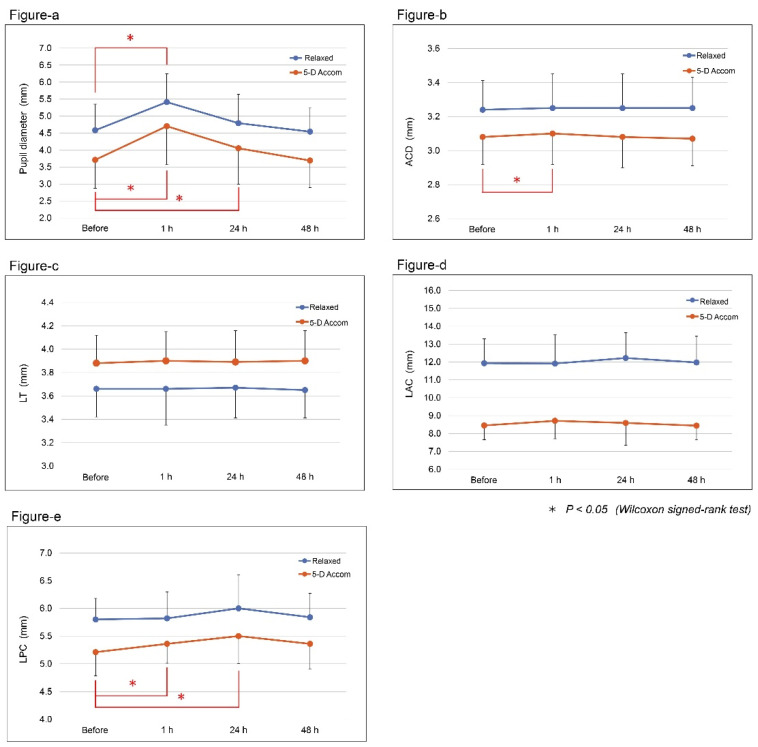
Change in biometric parameters after the instillation of 0.01% atropine eye drops. Shown is a comparison of the biometric parameters in the relaxed state (Relaxed) and with the 5 D accommodative stimulus (5-D Accom) before and 1, 24, and 48 h after instillation of 0.01% atropine eye drops. (**a**) The pupil diameter increased significantly 1 h after instillation in the relaxed state but returned to the pre-instillation level at 24 h. With the 5 D accommodative stimulus, the pupil diameter significantly increased at 1 and 24 h but returned to the pre-instillation level at 48 h. (**b**–**e**) In the non-accommodative eyes, none of the assessed biometric parameters (ACD, LT, LAC, LPC), with the exception of pupil diameter, showed changes at any time point when compared to that before the instillation. However, with the 5 D accommodative stimulus, ACD significantly increased at 1 h (**b**) and LPC significantly increased at 1 and 24 h (**e**) after atropine instillation when compared with that before instillation (*p* < 0.05). D, diopter; ACD, anterior chamber depth; LT, lens thickness; LAC, horizontal radius of the lens’ anterior surface curvature; LPC, horizontal radius of the lens’ posterior surface curvature. * *p* < 0.05, Wilcoxon signed-rank test.

**Table 1 jcm-10-04384-t001:** Baseline biometric parameters of both eyes before instillation of 0.01% atropine eye drops.

	Right Eye		Left Eye		*p*-Value *
	Median	IQR	Median	IQR	
Baseline biometric parameter					
Spherical equivalent (D)	−5.88	6.50	−5.37	6.94	0.69
Axial length (mm)	25.39	2.37	25.45	2.52	0.95
Accommodation amplitude (D)	6.48	1.49	6.53	1.07	0.95
Central corneal thickness (µm)	537	42	527	42	0.97
Anterior chamber depth (mm)	3.28	0.35	3.33	0.31	0.62
Pupil diameter (mm)	4.38	1.26	4.59	0.79	0.55
Lens thickness (mm)	3.61	0.25	3.61	0.27	0.96
Radius of the lens’ anterior surface curvature (mm)	11.65	1.97	12.04	3.39	0.57
Radius of the lens’ posterior surface curvature (mm)	5.73	0.60	5.81	0.67	0.86

IQR, interquartile range; D, diopters; * Mann–Whitney U test.

**Table 2 jcm-10-04384-t002:** Change in refraction, accommodation amplitude, and subjective symptoms after the instillation of 0.01% atropine eye drops.

	Pre-Instillation		1 h after Ocular Instillation		24 h after Ocular Instillation		48 h after Ocular Instillation	
	Median	IQR	Median	IQR	Median	IQR	Median	IQR
Spherical equivalent (D)	−5.88	6.50	−6.25	6.44	−6.13	6.13	−6.13	6.19
			(*p*, 0.10)		(*p*, 0.86)		(*p*, 0.55)	
Accommodation amplitude (D)	6.48	1.49	6.49	1.38	6.40	1.09	6.59	1.00
			(*p*, 0.76)		(*p*, 0.50)		(*p*, 0.07)	
Subjective symptoms	0.00	0.00	0.00	0.00	0.00	0.00	0.00	0.00
			(*p*, 0.11)					

D, diopters; h, hour; *p*, *p*-value (Wilcoxon signed-rank test).

## Data Availability

All data relevant to this study were included in the article.

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
