# Peer review of "Effects of 0.01% Atropine Instillation Assessed Using Swept-Source Anterior Segment Optical Coherence Tomography"

_jcm, 2021, doi:10.3390/jcm10194384_

Round 1
Reviewer 1 Report
This paper addresses an interesting topic but I have some remarks to be solved:
Line 28. Introduction: authors explain with detail the myopic progression process in the childhood but this study is in subjects aged 24-35 years; a paragraph should be added explaining what happens in this range of age.
Line 95. Participants: this study includes 17 participants from about -11D to 0D refractive errors; this issue should be discussed in the manuscript and how refractive status can affect the evaluated results.
Line 130. The participants’ accommodative amplitude was measured using the ARK-1 autorefractor; how these measurements were made should be explained.
Figure1. Caption should be completed clarifying the acronyms used: LPC, LT, LAC,….
Line 208 Discussion. The effect of atropine eye drops on accommodation and pupil diameter are evaluated but vergence are related to them, it should be added and discussed in the text or added as limitation of the study.
Correlations, if they exist, between subjective and objective results should be commented on the text.
Along the text, p values should be expressed as X.XX
Author Response
Response to Reviewer 1 Comments
We would like to thank you for the careful review of our manuscript and the helpful comments provided. The manuscript has been revised as suggested, and our point-by-point responses to each of the comments have been provided below.
Point 1:
Line 28. Introduction: authors explain with detail the myopic progression process in the childhood but this study is in subjects aged 24-35 years; a paragraph should be added explaining what happens in this range of age.
Response 1:
Thank you for your valuable suggestion. We agree with the comment. However, low-concentration atropine eyedrops for myopia have not approved in Japan. The IRB of our institute recommended us to adopt adults instead of children.
The following sentence was added to page 3 line 1 for clarity:
In this study, we examined young adults instead of the children, as low-concentration atropine eyedrops for myopia have not approved for children in Japan.
We have already written in limitation as follow below:
Our study has several limitations. First, the participants were young adults rather than school-aged children, which could have influenced the results. Low-concentration atropine instillation has been used to inhibit myopia progression in school- and preschool-aged children to address the trend of early-onset myopia and the increase in the number of preschool- and school-aged myopia patients. In this respect, the ocular permeability of atropine and its pharmacokinetics might differ between children and adults. The accommodation amplitude also differs between school-aged children and young adults. Hence, because the participants were young adults aged 24‒35 years, the results might not be directly applicable to school-aged children.
Point 2: Line 95. Participants: this study includes 17 participants from about -11D to 0D refractive errors; this issue should be discussed in the manuscript and how refractive status can affect the evaluated results.
Response 2:
Thank you for your valuable suggestion. The reviewer’s comment is correct.
Millodot.M reported that fundamental difference in the accommodative responses of myopes compared to emmetropes and hyperopes; myopes accommodated less.
However, in this study, we described that to ensure a participant’s ability to accommodate 5 D or greater, we also examined the accommodation of the participants using the ARK-1 autorefractor.
The following sentence was inserted in page 3 line 10 for clarity:
We considered the effect that degree of refractive error gives to accommodation factors, as 17 participants have from about -11D to 0D refractive errors. However in this study, to ensure a participant’s ability to accommodate 5 D or greater, we also examined the accommodation of the participants using the ARK-1 autorefractor.
We added a report of Millodot.M to the References [25]
Point 3: Line 130. The participants’ accommodative amplitude was measured using the ARK-1 autorefractor; how these measurements were made should be explained.
Response 3:
Thank you for your comment.
The following sentence has been added to page 3 line 42 for clarity:
Objective measurement of accommodation was performed with the participant focusing on a target, that moved to a near point from the distance.
Point 4: Figure1. Caption should be completed clarifying the acronyms used: LPC, LT, LAC,
Response 4:
Thank you for your suggestion, we have explained the acronyms used in the caption for Figure 1. These are as follows:
ACD: anterior chamber depth
LT: lens thickness
LAC: horizontal radii of the lens’ anterior surface curvature
LPC: horizontal radii of the lens’ posterior surface curvature
Point 5: Line 208 Discussion. The effect of atropine eye drops on accommodation and pupil diameter are evaluated but vergence are related to them, it should be added and discussed in the text or added as limitation of the study.
Correlations, if they exist, between subjective and objective results should be commented on the text.
Response 5:
We appreciate your valuable suggestion. In this study, we did not examine the effect of atropine eye drops on vergence.
As suggested, the following sentence has been added as one of the limitations of the study:
Although the effect of atropine eye drops on vergence reactions should have been evaluated, we did not examine this in the present study.
Point 6: Along the text, p values should be expressed as X.XX
Response 6: As suggested, we have revised the p values throughout the manuscript.
We thank you for the time and effort spent throughout the review process of this manuscript. We believe that the excellent comments provided by all did increase the utility of this paper. We hope we were able to revise the paper in accordance with the reviewers’ expectations. We are honored to be working on this paper with you throughout this revision process.

Reviewer 2 Report
Nice and important work, it could have clinical relevance in the future.
The authors performed non-parametric tests for comparison. The null hypothesis of these tests is that no differences are in median values of the samples. Instead of mean values, please use medians with interquartile range for the description.
Which method did you use for the measurement of axial length?
Author Response
Response to Reviewer 2 Comments
We would like to thank you for the careful review of our manuscript and the helpful comments provided. The manuscript has been revised according to the suggestions, and our point-by-point responses to each of the comments have been provided below.
Point 1: The authors performed non-parametric tests for comparison. The null hypothesis of these tests is that no differences are in median values of the samples. Instead of mean values, please use medians with interquartile range for the description.
Response 1: Thank you for your valuable suggestion. we have revised Tables 1 and 2 using medians with interquartile ranges. However, we did not make any changes to Figure 1 to facilitate comparison with other reports in the discussion.
Point 2: Which method did you use for the measurement of axial length?
Response 2: Thank you for your comment. We conducted the measurement of axial length using optical axial length measuring device (OPTICAL BIOMETER OA-2000, Tomey Corp., Nagoya, Japan)
The following sentence was added in page 3 line 43 for clarity:
And we conducted the measurement of axial length using optical axial length measuring device (OPTICAL BIOMETER OA-2000, Tomey Corp., Nagoya, Japan)
We thank you for the time and effort spent throughout the review process of this manuscript. We believe that the excellent comments provided by all did increase the utility of this paper. We hope we were able to revise the paper in accordance with the reviewers’ expectations. We are honored to be working on this paper with you throughout this revision process.

Round 2
Reviewer 1 Report
No